# Tolerability and efficacy of switching anti-fibrotic treatment from nintedanib to pirfenidone for idiopathic pulmonary fibrosis

**Keishi Sugino\***, **Hirotaka Ono, Mikako Saito, Masahiro Ando, Eiyasu Tsuboi**

Department of Respiratory Medicine, Tsuboi Hospital, Koriyama city, Fukushima, Japan

\* ks115108@tsuboi-hp.or.jp

**Data Availability Statement:** The datasets used and/or analyzed during the current study are available from supporting information files.

## Abstract

### Background

In real-world studies, the rate of discontinuation of nintedanib (NT) varies from 4% to 53%. Switching anti-fibrotic treatment in patients with idiopathic pulmonary fibrosis (IPF) has not been adequately investigated, and data on the tolerability and efficacy of changes in anti-fibrotic treatment is limited in clinical practice.

### Objective

To identify factors associated with poor continuation of NT, efficacy and predictors of deterioration after switching from NT to pirfenidone (PFD) in patients with IPF.

### Subjects and methods

One hundred and seventy patients with IPF in whom NT was introduced between April 2017 and March 2022 were included to investigate NT continuation status and the effect of switching to PFD.

### Results

A total of 123 patients (72.4%) continued NT for 1 year and had a significantly higher % forced vital capacity (FVC) at NT introduction than those who discontinued within 1 year (80.9% ± 16.3% vs. 71.9% ± 22.1%, P = 0.004). The determinant of poor NT continuation was the high GAP stage. On the other hand, 28 of 36 patients who discontinued NT because of disease progression switched to PFD. Consequently, FVC decline was suppressed before and after the change. The predictor of deterioration after the switch was a lower body mass index.

### Conclusions

In patients with IPF, early NT introduction increased continuation rates, and switching to PFD was effective when patients deteriorated despite initial NT treatment.

**Funding:** The author(s) received no specific funding for this work.

**Competing interests:** All authors contributed substantially to this work and are responsible for the consent of the manuscript. KS have received lecture fees from Nippon Boehringer Ingelheim Co., Ltd and Shionogi & Co., Ltd. The other authors have no financial relationships relevant to this article. All authors have no competing non-financial interests.

**Abbreviations:** IPF, idiopathic pulmonary fibrosis; NT, nintedanib; PFD, pirfenidone; HRCT, high-resolution computed tomography; AE, acute exacerbation; BMI, body mass index; KL-6, Krebs von den Lungen–6; SP-D, surfactant protein–D; 6MWT, 6-minute walking test; 6MWD, 6-minute walking distance; GAP, gender–age–physiology; FVC, forced vital capacity; %FVC, percentage predicted FVC; DLco, diffusing capacity of the lung for carbon monoxide; PFT, pulmonary function test; PaO2, partial pressure of oxygen in arterial blood; CI, confidence interval; SD, standard deviation; PH, pulmonary hypertension.

## Introduction

Idiopathic pulmonary fibrosis (IPF) has a chronic and progressive course, eventually leading to irreversible honeycomb lung formation, with a median survival duration of 2–3.5 years [1,2]. Moreover, because the clinical course is heterogeneous, the prognosis of IPF is difficult to predict [3,4]. Therefore, the timing of treatment with anti-fibrotic agents may not be easy in real-world clinical practice.

Anti-fibrotic agents including nintedanib (NT) and pirfenidone (PFD), have been recommended for the first-line treatment of IPF in international clinical practice guidelines [5]. IPF registry studies worldwide have reported significantly longer transplant-free survival in the anti-fibrotic drug-using group than in the nonuser group [6–8]. Thus far, the goal of IPF treatment is to stabilise or delay disease progression. A post—hoc analysis of the INPULSIS study [9] revealed that NT was as effective in preventing the annual decline in forced vital capacity (FVC) in mild cases with low gender—age—physiology (GAP) stages (GAP stages 1 and 2) as in severe cases (GAP stages 3 and 4). In addition, Fletcher et al. reported that increasing age and decreasing FVC at pre-treatment are associated with an increased probability of discontinuation during 52 weeks of NT treatment [10]. Although several studies reported that the rate of discontinuation of NT varies from 4% to 53%, a two-centre retrospective study demonstrated that patients with IPF who were referred to interstitial lung disease centres at an early stage had a higher rate of continuation of anti-fibrotic drugs and a better prognosis [11].

Because PFD and NT are characterised by different mechanisms of action and pharmacological profiles, switching anti-fibrotic drugs can be required in clinical practice when drug intolerance or disease progression despite of anti-fibrotic treatment is recognised. A few studies with small sample sizes demonstrated that second-line NT treatment could be effective for patients with IPF who have discontinued PFD treatment [12–17]. However, some patients with IPF who were switched from PFD to NT showed rapid disease progression or early termination despite treatment [12,13]. To the best of our knowledge, there is currently no evidence for the feasibility and efficacy of second-line PFD treatment in patients with IPF who discontinued their previous first-line NT treatment.

Thus, this study aimed to identify factors associated with NT discontinuation, efficacy, and predictors of deterioration after switching from NT to PFD in patients with IPF.

## Methods

### Study population

This retrospective study was conducted on 196 consecutive patients with IPF who were treated with NT as the first-line anti-fibrotic drugs at our hospital from April 2017 to March 2022 using the medical records of all subjects. The NT administration status was evaluated in April 2023 based on a chart review. Twenty—six patients with IPF were excluded from this study for the following reasons: missing data at follow-up (n = 9), the absence of pulmonary function test results at baseline (n = 8), the presence of the complication of lung cancer at the initial visit (n = 6) and the observation period being less than 6 months (n = 3). A total of 170 patients were included. Patients who were able to receive NT for more than 12 months were assigned to the continued NT group (n = 123); on the other hand, patients who were unable to receive NT over 12 months were assigned to the discontinued NT group (n = 47). All patients with IPF in our hospital have been started on treatment with NT at an initial dose of 300 mg. During the observation period, 130 patients discontinued NT for the following reasons: disease progression (n = 36), adverse effects (n = 53), death (n = 30), self-interrupted (n = 4), lung transplantation (n = 3) and other reasons (n = 4). In addition, 28 of 36 patients were switched

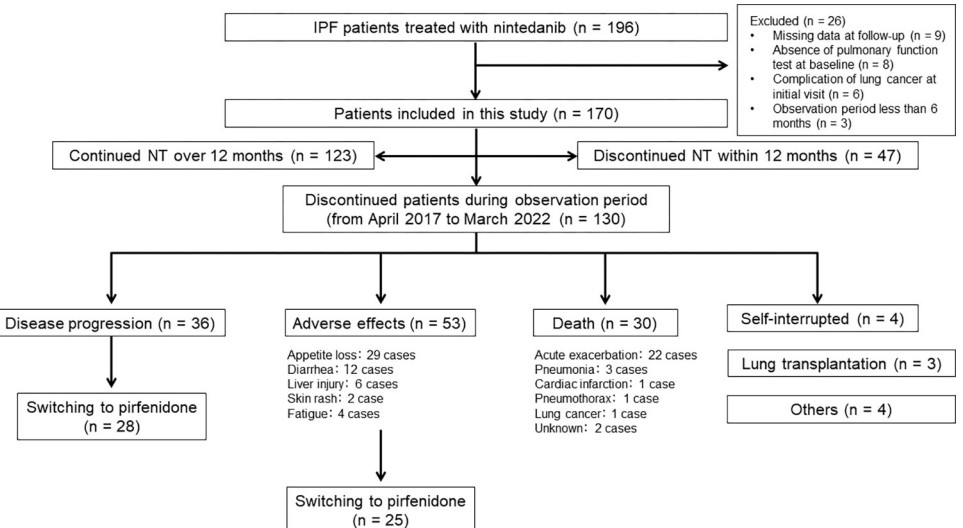

**Fig 1. Flowchart of the participant selection process.** In total, 196 IPF patients who had been treated with nintedanib (NT) were enrolled from April 2017 to March 2022. Patients with missing data at follow-up (n = 9), the absence of pulmonary function test results at baseline (n = 8), the presence of complications of lung cancer at the initial visit (n = 6), and an observation period of less than six months (n = 3) were excluded. Of the 170 patients, 123 were classified as continuing NT for more than 12 months and 47 were classified as discontinuing NT within 12 months. During the observation period, 130 patients discontinued NT as a first-line treatment. Their reasons for discontinuing the treatment were as follows: Disease progression (n = 36), adverse events (n = 53), death (n = 30), self-interrupted (n = 4), lung transplantation (n = 3), and others (n = 4). In addition, 28 of 36 patients were switched from NT to pirfenidone due to disease progression and 25 of 53 due to adverse effects.

from NT to pirfenidone because of disease progression and 25 of 53 were switched because of adverse effects (Fig 1).

IPF was diagnosed based on the recent American Thoracic Society (ATS)/European Respiratory Society (ERS)/Japanese Respiratory Society (JRS)/Latin American Thoracic Society (ALAT) international IPF guidelines [1,2]. The diagnosis of all patients was evaluated by a multidisciplinary team based on clinical, radiological and/or pathological findings.

The acute exacerbation (AE) of IPF was diagnosed per the criteria proposed by Collard et al. [18], and all of the following four conditions must be satisfied: i) previous or concurrent diagnosis of IPF; ii) acute worsening or the development of dyspnoea typically of <-1 month's duration; iii) CT with new bilateral ground-glass opacity and/or consolidation superimposed on a background pattern consistent with the UIP pattern; iv) deterioration not fully explained by cardiac failure or fluid overload.

The GAP score was calculated by sex, age, predicted FVC% and predicted diffusing capacity of the lung for carbon monoxide (DLco)%. Patients were divided according to the severity of staging into stages I—III as previously described [19].

To evaluate treatment response, we defined disease progression that met at least one of the following progression criteria within six months: i) a relative decline of ≥10% of the predicted FVC (%FVC); ii) a relative decline of ≥15% of the predicted DLco (%DLco); iii) a relative decline of ≥5% but <10% decline in %FVC along with increasing fibrosis in chest CT images; iv) a relative decline of ≥5% but <10% decline in %FVC along with deteriorating respiratory symptoms [20,21].

This study was approved by the Institutional Review Board of Tsuboi Hospital (no. Zizan 3–4). The Ethical Committees of Tsuboi Hospital waived the need for informed consent

because of the retrospective nature of the study. The study was conducted per the principles of the Declaration of Helsinki.

## Chest CT scan

Chest CT scans were performed using helical CT scanners (Aquilion 16, Toshiba, Tokyo, Japan and Aquilion Prime, Canon, Tokyo, Japan). Thin-section CT scans were obtained at full inspiration, and the scanning protocol consisted of the reconstruction of 1-mm slice thicknesses with a high-spatial-frequency algorithm. Thin-section CT images of the chest were photographed at window settings appropriate for the lung parenchyma (the window level ranging from—600 to—450 Hounsfield Units [HU] and the width ranging from 1600 to 1900 HU) for all patients. A consensus reading of the CT images was independently analysed by two pulmonologists (K.S., H.O.) and one radiologist (A.K.).

## Pulmonary function test

Spirometry and the DLco measurements were performed using a pulmonary function test (PFT) system (Chestac-8900, Chestac-8900α, CHEST Co. Ltd., Tokyo, Japan). The diffusion capacity was measured using the single-breath technique. These PFTs were performed by two technicians using the method described in the ATS criteria [22].

## Doppler echocardiography

The estimated systolic pulmonary arterial pressure (esPAP) was calculated from measurements using transthoracic Doppler echocardiography by specific technicians. The transtricuspid pressure gradient was calculated using the modified Bernoulli equation and was considered to be equal to the equal to the esPAP in the absence of right ventricular outflow obstruction: esPAP = transtricuspid pressure gradient + right atrial pressure. Pulmonary arterial hypertension was defined as a esPAP >35 mmHg at rest [23].

## Measurement of the levels of the serum markers

Serum Krebs von den Lungen-6 (KL-6) and surfactant protein-D (SP-D) levels were measured using peripheral blood samples collected from patients at initial admission to our hospital. The serum level of KL-6 (normal < 500 U/ml) was measured via enzyme-linked immunosorbent assay (ELISA) using the ELTEST KL-6 kit (Eiken chemical, Tokyo, Japan), and that of SP-D (normal < 110 ng/ml) was measured using a commercial ELISA kit (Yamasa, Tokyo, Japan).

## Statistical analysis

Continuous data are expressed as the mean ± standard deviation (SD) and categorical data as frequencies and percentages in each group. Continuous data comparisons between the two groups were performed using the Wilcoxon rank-sum test or Student's *t* test as appropriate. The chi-square test or Fisher's exact test (as appropriate) was used to compare categorical variables. Mean changes in FVC and body weight values for six months before and after treatment with anti-fibrotic agents were compared using the one-way repeated-measure analysis of variance (ANOVA) with Bonferroni's multiple comparison test. Univariate and multivariate logistic regression analyses evaluated clinical features associated with NT discontinuation and predictors of deterioration after the switch. Variables identified by the univariate analysis ($p < 0.05$) were evaluated by the multivariate analysis. To avoid multi-collinearity, only one of the highly correlated variables (Pearson's correlation coefficient $\geq 0.7$) was entered in the multivariate mode. The optimal cut-off value of the body mass index (BMI) threshold for

**Table 1. Comparison of the characteristics of IPF patients continuing and discontinuing NT.**

| Variable | Continued NT ≥12 months (n = 123) | Discontinued NT <12 months (n = 47) | P-value |
|---|---|---|---|
| Age, yrs | 73.9 ± 7.9 | 73.4 ± 7.4 | 0.78 |
| Sex, male/female | 103/20 | 30/17 | 0.007 |
| Body weight (kg) | 62.1 ± 11.8 | 54.5 ± 13.3 | 0.0004 |
| Body mass index (kg/m$^2$) | 24.0 ± 3.7 | 21.9 ± 4.0 | 0.001 |
| PaO$_2$ (Torr) | 76.9 ± 11.5 | 76.8 ± 12.7 | 0.96 |
| 6MWD, m | 419 ± 112 | 381 ± 151 | 0.08 |
| 6MWT, Lowest SpO$_2$, % | 85.2 ± 5.8 | 83.6 ± 6.9 | 0.14 |
| FVC % predicted | 80.9 ± 16.3 | 71.9 ± 22.1 | 0.004 |
| DLco % predicted | 69.9 ± 21.7 | 66.1 ± 24.0 | 0.38 |
| PH (>35 mmHg) | 13 (10.5) | 7 (14.8) | 0.43 |
| GAP stage (stage I/II/III) | 80/34/9 | 24/11/12 | 0.005 |
| NT dose reduction | 35 (28.4) | 21 (44.6) | 0.06 |
| Time since diagnosis months | 23.6 ± 19.7 | 23.2 ± 18.4 | 0.91 |

Data are presented as mean ± SD or n (%). IPF: Idiopathic pulmonary fibrosis, NT: Nintedanib, PaO$_2$: Partial pressure of arterial oxygen, 6MWD: 6-minute walking distance, 6MWT: 6-minute walking test, FVC: Forced vital capacity, DLco: Diffusion capacity of the lung for carbon monoxide, PH: Pulmonary hypertension, GAP: Gender, age, and lung physiology.

predicting NT discontinuation and deterioration after the switch was derived from the receiver operating characteristic (ROC) curve. The threshold for statistical significance was set at p< 0.05. Data analyses were performed using JMP software (version 14.2.0, SAS Institute Inc., Cary, NC, USA) and SPSS Statistics for windows (IBM Corp. Released 2016. IBM SPSS Statistics for Windows, Version 24.0. Armonk, NY: IBM Corp).

## Results

### Comparison of characteristics of patients with IPF continuing and discontinuing NT

Among 170 patients with IPF, 123 (72.4%) were able to receive NT for ≥12 months, while 47 others discontinued NT for <12 months. The ratio of males, baseline body weight, BMI, and %FVC were significantly higher in patients who continued NT than in those who discontinued it. The baseline disease severity (GAP stage) was significantly milder in patients who continued NT than in those who discontinued it (Table 1).

Of note, 123 patients (72.4%) were able to continue NT for over 12 months. Forty-three (35%) of these patients continued without interruption or dose reduction. Next, 80 patients (65%) had to reduce the NT dose during 12 months because of the following reasons: anorexia/nausea (n = 31, 39%), diarrhoea (n = 23, 29%), liver injury (n = 20, 25%) and fatigue (n = 6, 7%). On the other hand, 47 patients had to discontinue NT treatment within 12 months because of adverse effects (n = 25, 53%), death (n = 15, 32%), or worsening of their current disease (n = 7, 15%) (Table 2).

### Comparison of the tolerability of IPF patients to NT treatment between elderly and non-elderly groups

There was no difference in the time to dose reduction between elderly (≥75 years or ≥80 years) and non-elderly (<75 years or <80 years) patients with IPF. The percentage of NT dose

**Table 2. Treatment status of NT.**

| Variable | |
|---|---|
| **Continued NT ≥12 months** | N = 123 |
| Continued without interruption and/or dose reduction | 43 (35) |
| Interruption and/or dose reduction | 80 (65) |
| Reason for interruption and/or dose reduction | |
| Anorexia/nausea | 31 (39) |
| Diarrhoea | 23 (29) |
| Liver injury | 20 (25) |
| Fatigue | 6 (8) |
| **Discontinued NT <12 months** | N = 47 |
| Reason for discontinuation | |
| Adverse effects | 25 (53) |
| Anorexia/nausea | 12 (48) |
| Diarrhoea | 4 (16) |
| Liver injury | 4 (16) |
| Fatigue | 3 (12) |
| Skin rash | 2 (8) |
| Death | 15 (32) |
| Disease progression | 7 (15) |

Data are presented as n (%). NT: Nintedanib.

reduction within six months was significantly higher in the elderly group (≥75 years vs. <75 years; 54.6% vs. 39.2%, P = 0.04, ≥80 years vs. <80 years; 60.8% vs. 41.9%, P = 0.03), whereas the rate of NT dose reduction within 12 months did not differ significantly between the two age groups. There was no difference in the duration of NT discontinuation between the two groups. In addition, no significant difference was observed in the NT discontinuation rate between elderly and non-elderly patients with IPF during the six-month and twelve-month periods. However, NT was discontinued for significantly more reasons due to adverse effects in the elderly group than in the non-elderly group (Table 3).

## Baseline %FVC of patients continuing NT for 12 months (stratified by % FVC)

The stratified analysis for %FVC of patients continuing NT for 12 months demonstrated that 2 (18.2%) of them had a baseline predicted FVC <50%, 9 (56.2%) of them had a predicted FVC between ≥50% and <60%, 19 (73.0%) had a predicted FVC between ≥60% and <70%, 31 (72.1%) had a predicted FVC between ≥70% and <80% and 62 (83.7%) of them had a predicted FVC ≥80% (Fig 2).

## Risk factors for NT discontinuation within 12 months

With regard to risk factors for NT discontinuation within 12 months, univariate logistic regression analyses showed that the risk factors were male sex, decreased %FVC and BMI and increased GAP staging. Regarding the most unfavourable predictive factors for discontinuing NT within 12 months, the multivariate logistic regression analysis revealed the following factors; GAP stage 3 against stage 1 and GAP stage 3 against stage 2 (OR = 5.031, 95%CI: 1.721–15.398, P = 0.003, OR = 3.263, 95%CI: 1.010–10.970, P = 0.04) (Table 4).

**Table 3. Comparison of the tolerability of IPF patients to NT treatment between elderly and non-elderly groups.**

| Variable | Patients ≥75 yrs | Patients <75 yrs | *P- value | Patients ≥80 yrs | Patients <80 yrs | **P- value |
|---|---|---|---|---|---|---|
| Duration of dose reduction, days | 116±127 | 167±181 | 0.08 | 115±132 | 151±166 | 0.26 |
| Dose reduction during 6 M | 47 (54.6) | 33 (39.2) | 0.04 | 28 (60.8) | 52 (41.9) | 0.03 |
| Dose reduction during 12 M | 56 (65.1) | 51 (60.7) | 0.63 | 30 (65.2) | 77 (62.1) | 0.85 |
| Causes of dose reduction | | | | | | |
| AEs | 60 (69.7) | 55 (65.4) | 0.62 | 33 (71.7) | 82 (66.1) | 0.58 |
| Duration of discontinuation, days | 485±332 | 451±316 | 0.55 | 507±315 | 453±328 | 0.38 |
| Discontinuation during 6 M | 15 (17.4) | 10 (11.9) | 0.38 | 7 (15.2) | 18 (14.5) | 1.00 |
| Discontinuation during 12 M | 23 (26.7) | 24 (28.5) | 0.86 | 11 (23.9) | 36 (29.0) | 0.56 |
| Causes of discontinuation | | | | | | |
| AEs | 36 (41.8) | 17 (20.2) | 0.002 | 20 (43.4) | 33 (26.6) | 0.03 |
| Disease progression | 19 (22.0) | 20 (23.8) | 0.79 | 9 (19.5) | 30 (24.1) | 0.52 |
| Death | 15 (17.4) | 15 (17.8) | 0.94 | 9 (19.5) | 21 (16.9) | 0.68 |
| Self-interrupted | 2 (2.3) | 2 (2.3) | 0.98 | 1 (2.1) | 3 (2.4) | 0.92 |
| Cancer | 1 (1.1) | 2 (2.3) | 0.61 | 1 (2.1) | 2 (1.6) | 1.00 |
| Others | 0 (0) | 1 (1.1) | 0.49 | 0 (0) | 1 (0.8) | 1.00 |

Data are presented as n (%). IPF: Idiopathic pulmonary fibrosis, NT: Nintedanib, yrs: Years, M: Months, AEs: Adverse effects.

* Patients ≥75 years vs. <75 years.

** Patients ≥80 years vs. <80 years.

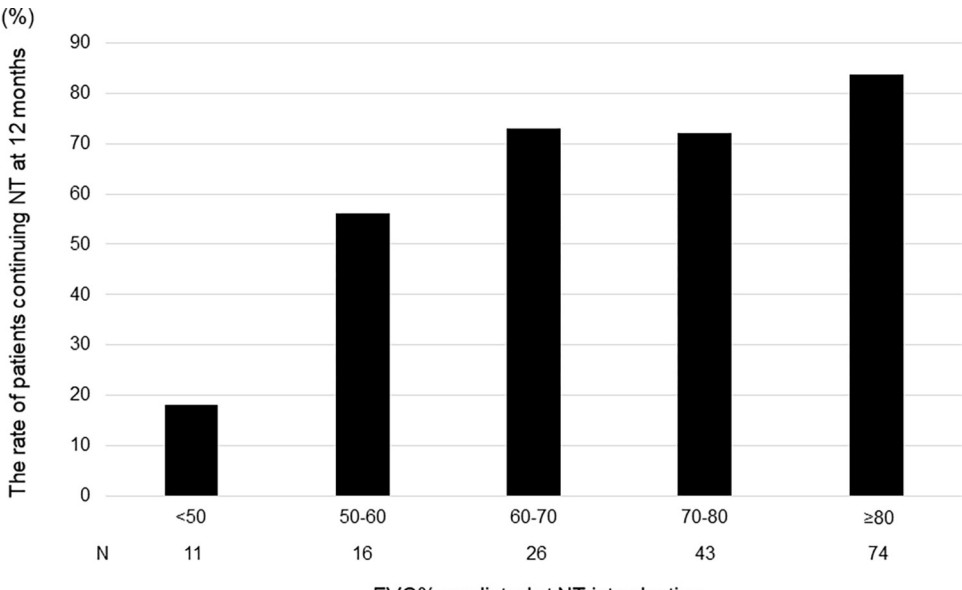

**Fig 2. The predicted baseline forced vital capacity % of patients continuing nintedanib during 12 months (stratified by FVC % predicted).** The stratified analysis for %FVC of patients continuing NT for 12 months demonstrated that 2 (18.2%) of them had a baseline predicted FVC <50%, 9 (56.2%) had a predicted FVC between ≥50% and <60%, 19 (73.0%) had a predicted FVC between ≥60% and <70%, 31 (72.1%) had a predicted FVC between ≥70% and <80%, and 62 (83.7%) had a predicted FVC of ≥80%.

**Table 4. Risk factors for discontinuing NT within 12 months.**

| Variable | Odds ratio (95% CI) | P-value |
|---|---|---|
| **Univariate analysis** | | |
| Age, yrs | 1.005 (0.963–1.049) | 0.79 |
| Male sex | 0.342 (0.159–0.735) | 0.006 |
| Body mass index (kg/m$^2$) | 0.863 (0.786–0.947) | 0.001 |
| FVC % predicted | 0.966 (0.946–0.986) | 0.0006 |
| DLco % predicted | 0.992 (0.974–1.009) | 0.37 |
| GAP stage 3 vs. stage 1 | 7.312 (2.720–19.654) | <0.0001 |
| GAP stage 3 vs. stage 2 | 5.781 (1.968–16.975) | 0.001 |
| **Multivariate analysis** | | |
| GAP stage 3 vs. stage 1 | 5.031 (1.721–15.398) | 0.003 |
| GAP stage 3 vs. stage 2 | 3.263 (1.010–10.970) | 0.04 |

NT: Nintedanib, CI: Confidence interval, FVC; forced vital capacity, DLco; diffusing capacity for carbon monoxide, GAP: Gender, age, and lung physiology.

## Efficacy of PFD in patients who switched from NT

Fifty-three (31.1%) of 170 patients were switched from NT to PFD because of disease progression (n = 28) and intolerable adverse effects (n = 25) (Fig 1). Of the 25 patients who were switched from NT to PFD because of adverse effects, six were discontinued during the study (three due to adverse effects, two due to death and one due to self-interruption). After switching from NT to PFD because of disease progression, 19 patients had stable disease and nine others had worsening disease. The reasons for worsening after the switch were as follows: in 4 patients, %FVC and %DLco decreased, symptoms worsened, and imaging findings worsened; in 3 patients, %FVC decreased, symptoms worsened, and imaging findings worsened; and in 2 patients, %DLco decreased, symptoms worsened, and imaging findings worsened. Regarding changes in FVC before and after treatment with switching antifibrotic drugs, the mean FVC value was 2.46 ± 0.52 L under NT treatment six months prior to the switch to pirfenidone, 2.16 ± 0.50 L at the time of the switch, and 2.08 ± 0.55 L six months after PFD initiation. The decline in FVC was suppressed after switching to PFD as demonstrated using the one-way repeated-measure ANOVA with Bonferroni's multiple comparison (P = 0.0001, P = 0.3) (Fig 3). Also, the weight change in the 6 months before PFD switch was -0.8±3.8 kg and in the 6 months after PFD switch was -1.4±3.3 kg, showing no difference in weight change before and after PFD switch (64.6 ± 10.3 kg, 63.7 ± 9.9 kg, 62.3 ± 11.2 kg; P = 0.69, P = 0.16). On the other hand, patients who switched to PFD due to NT's adverse effects also had stable FVC values six months after the switch (2.52 ± 0.49 L, 2.45 ± 0.46 L, 2.38 ± 0.54 L; P = 0.06).

The clinical course after the switch from NT to PFD because of the deterioration of the current disease is shown in Fig 4 as swimmer plots. The mean (median) periods for NT and PFD administration were 356 ± 242 (324) days, and 705 ± 305 (673) days, respectively. Thirteen of the 28 patients discontinued treatment after the change for the following reasons: death (11 cases), lung transplantation (one case) and adverse effects such as liver injury (one case). The causes of death were respiratory failure due to worsening of the present illness in five patients, AE in four patients, pneumonia in one patient, and lung cancer in one patient (Fig 4).

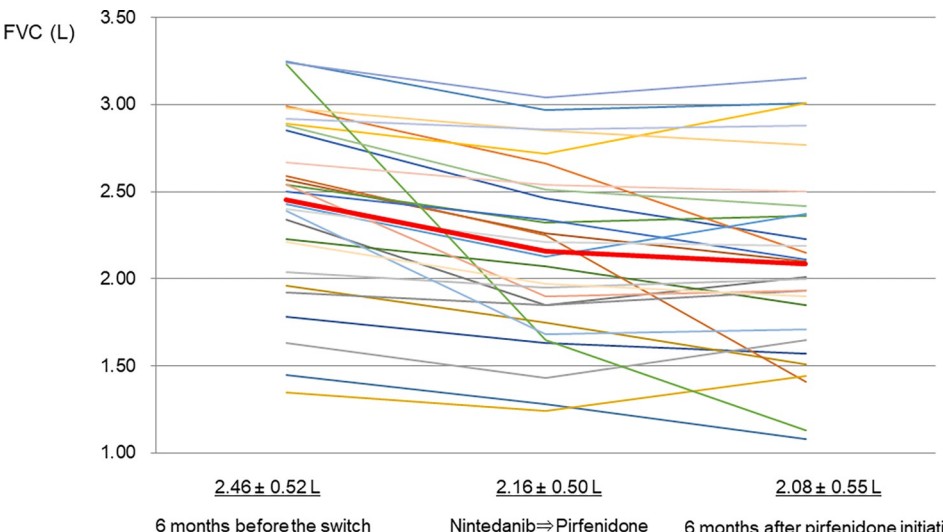

**Fig 3. Change in FVC before and after treatment with switching anti-fibrotic drugs.** Each line represents one patient. The bold red line represents the mean value. The mean FVC value was 2.46 ± 0.52 L under nintedanib treatment six months prior to switch to pirfenidone, 2.16 ± 0.50 L at the time of the switch, and 2.08 ± 0.55 L at six months after pirfenidone initiation. The decline in FVC was suppressed after switching to pirfenidone using the one-way repeated-measures ANOVA with Bonferroni's multiple comparison (P = 0.0001, P = 0.3).

## Predictors of deterioration after switching from NT to PFD

Univariate logistic regression analyses of predictors of deterioration after switching from NT to PFD revealed that lower BMI and %FVC values at the time of the switch were significant covariates (OR = 0.514, 95%CI: 0.269–0.803, P = 0.001, OR = 0.944, 95%CI: 0.872–0.998, P = 0.04). In multivariate logistic regression analyses, a lower BMI was the most significant predictor of deterioration (OR = 0.511, 95%CI: 0.289–0.903, P = 0.003) (Table 5).

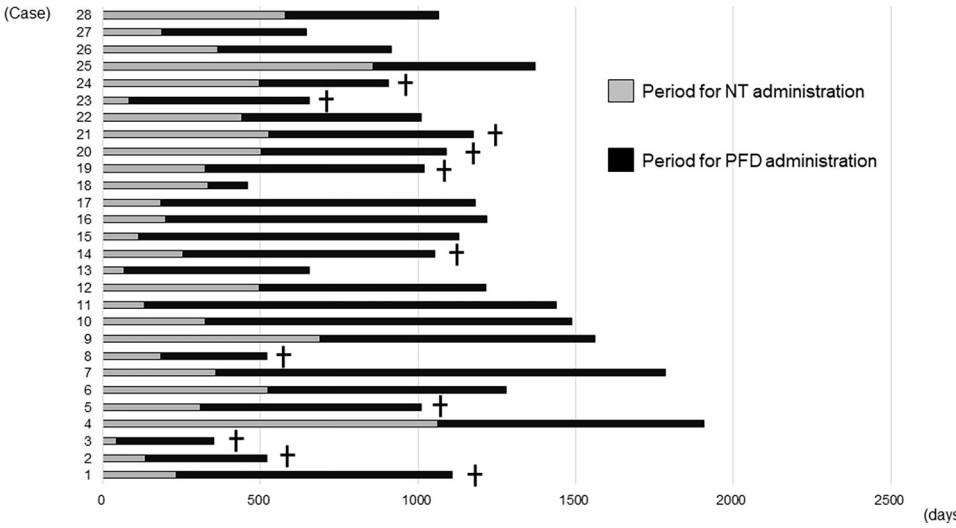

**Fig 4. Swimmer plots showing the time course for switching from nintedanib to pirfenidone.** The mean (median) periods for NT and PFD administration were 356 ± 242 (324) days, and 705 ± 305 (673) days, respectively. Thirteen of the 28 patients discontinued the treatment after the switch. The reasons for discontinuation were as follows: [Died: 11 (cases no. 1, 2, 3, 5, 8, 14, 19, 20, 21, 23, 24), Lung transplantation: 1 (case no. 18), Liver injury (as an adverse effect): 1 (case no. 13)]. The causes of death were respiratory failure due to the worsening of the present illness in five patients, AE in four patients, pneumonia in one patient, and lung cancer in one patient.

**Table 5. Logistic regression analysis for verifying predictors of deterioration after switching from NT to PFD.**

| Variable | Odds ratio (95% CI) | *P*-value |
|---|---|---|
| **Univariate analysis** | | |
| Age, yrs | 1.027 (0.887–1.189) | 0.71 |
| Male sex | 2.133 (0.257–45.541) | 0.50 |
| Body weight (kg) | 0.924 (0.826–1.009) | 0.08 |
| Body mass index (kg/m$^2$) | 0.514 (0.269–0.803) | 0.001 |
| FVC % predicted | 0.944 (0.872–0.998) | 0.04 |
| DLco % predicted | 0.997 (0.951–1.046) | 0.91 |
| GAP stage 3 vs. stage 1 | 12 (0.897–160.403) | 0.06 |
| GAP stage 3 vs. stage 2 | 6 (0.422–85.247) | 0.18 |
| **Multivariate analysis** | | |
| Body mass index (kg/m$^2$) | 0.511 (0.289–0.903) | 0.003 |
| FVC % predicted | 0.952 (0.889–1.019) | 0.08 |

NT: Nintedanib, PFD: Pirfenidone, CI: Confidence interval, FVC; forced vital capacity, DLco; diffusing capacity for carbon monoxide, GAP: Gender, age, and lung physiology.

Moreover, in ROC analyses, a BMI threshold of 23.3 was identified as the greatest predictor of deterioration after the anti-fibrotic drugs switch (AUC; 0.818, P = 0.001, sensitivity; 77.8%, specificity; 84.2%, positive predictive value; 70.0%, negative predictive value; 88.9%).

## Discussion

These results indicate that the early introduction of NT increases continuation rates in patients with IPF and that switching to PFD is effective when the conditions of patients deteriorate despite the initial NT treatment.

IPF has a chronic and progressive clinical course that eventually leads to the irreversible formation honeycombed lungs, with restricted ventilation and impaired lung diffusion on respiratory function tests [1,2]. Patients may present with slow progression over several years to several decades or more, rapid progression, or even progressive deterioration with repeated exacerbations (including AE) during the disease course [3]. Because of this diversity in disease behaviour, the appropriate diagnosis and therapeutic intervention are not easy in general practice.

In patients with IPF, the anti-fibrotic drugs (NT and PFD) are currently the first-line drugs [5]. IPF registry studies around the world have shown significantly longer transplant-free survival in the anti-fibrotic drugs use group than in the non-use group [6–8]. A post—hoc analysis of the INPULSIS study showed that NT was as effective in mild cases with low GAP stages (GAP stages 1 and 2) as in severe cases with high GAP stages (GAP stages 3 and 4) in preventing the annual decline in FVC, indicating that NT is also effective in patients with mild IPF [9]. We also reported the effectiveness of early intervention with anti-fibrotic agents in patients with IPF and mild disease based on the Japanese severity classification [24]. Several studies have reported that the rate of NT discontinuation varies from 4% to 53% [25]. Recently, an interim report of a post-marketing surveillance for NT in Japan demonstrated that 2795 (50.1%) of 5578 patients with IPF discontinued NT within 12 months [26]. Diarrhoea (13.2%) was the most common reason for NT discontinuation within 12 months [26]. In our study, 47 (28.7%) of 170 patients with IPF discontinued NT within 12 months, and the most common adverse effect associated with NT discontinuation within 12 months was anorexia/nausea (48%). The high continuation rate of NT in our study can be attributed to the fact that it was

often administered to mild cases as the first-line treatment. Although diarrhoea was the most common reason for NT discontinuation within 12 months because of adverse effects according to this report [26], the sum of decreased appetite, nausea and vomiting exceeded diarrhoea, indicating that their results were similar to those of our study. Moreover, Kato et al. reported that anorexia/nausea was the most common adverse effect in the non-continuation group, and the frequency of anorexia/nausea was significantly higher in the non-continuing group than in the NT continuation group [27].

Brereton et al. reported that patients with IPF who are referred to ILD centres at an early stage have a higher rate of continuation of anti-fibrotic therapy and a better prognosis [11]. Fletcher SV et al. reported that increasing age and decreasing FVC at pre-treatment is associated with an increased probability of discontinuation during 52 weeks of NT treatment [10]. In addition, they reported a high NT discontinuation rate of 76.5% identified in the subgroup with %FVC <50% [10]. In our study, %FVC was significantly higher in patients who continued NT than in those who discontinued it, and patients with %FVC <50% also had a high rate of NT discontinuation (81.8%), almost twice as high as that in patients with %FVC 50–60% (43.8%). Moreover, the GAP stage was significantly lower in patients who continued NT than in those who discontinued it. Recently, Oishi et al. reported that as a patient's performance status (PS) worsened, more anti-fibrotic drugs were discontinued in patients with IPF, and a significant negative correlation was found between worsening PS and %FVC [28]. Therefore, we believe that NT treatment should be introduced early for patients with IPF in terms of the continuation rate of anti-fibrotic agents.

The ATS/ERS/JRS/ALAT international guideline for IPF recommends the use of NT in patients with IPF [5]. However, the age of the patient is not considered the guidelines, and there is a paucity of data on the tolerability and safety of NT in elderly patients with IPF in clinical practice. Cilli A et al. reported that elderly patients with IPF (≥75 years) had significantly more adverse events and dose reductions with the use of anti-fibrotic drugs than non-elderly patients (<75 years), whereas the discontinuation rates of anti-fibrotic drugs were similar in both age groups [29]. Hirari et al. also stated that the need for NT dose reduction was significantly higher in patients with IPF aged 80 years and older (50% vs. 26.8%, P = 0.039) [30]. On the other hand, Uchida Y, et al. reported that the incidence of NT treatment discontinuation was higher in elderly (≥75 years) than in non-elderly (<75 years) patients with IPF [31]. Furthermore, they demonstrated that early NT treatment termination within six months resulted in lower BMI and %FVC values [31]. A post-marketing surveillance study of Japanese IPF patients treated with nintedanib showed that older age (≥75 years), male sex, smaller body surface area (1.58 m$^2$), severe IPF, and lower predicted %FVC (< 70%) at baseline were associated with an increased risk of early discontinuation of treatment with NT [26]. In our study, the NT discontinuation rates did not differ significantly between elderly (≥75 years or ≥80 years) and non-elderly (<75 years or <80 years) patients with IPF. Risk factors for discontinuation within 12 months were higher GAP severity, but older age was not a significant factor. We believe this was partly due to the subjective judgement of the attending physician regarding the elderly. As a result, the proportion of patients who reduced their NT dose earlier within 6 months was significantly higher in elderly patients than in the non-elderly patients. In addition, this may be due to the fact that the %FVC (≥75 years: 76.6±17.9%, ≥80 years: 77.8±17.6%), %DLco (≥75 years: 73.9±23.8%, ≥80 years: 74.2±25.9%), and BMI (≥75 years: 22.6±3.9%, ≥80 years: 22.1 ±4.4%) were relatively preserved in elderly patients.

Because PFD and NT have different mechanisms of action and pharmacological profiles, switching anti-fibrotic drugs can be required in clinical practice in cases of drug intolerance or disease progression despite of anti-fibrotic treatment. Several studies have reported on the tolerability and efficacy of second-line NT treatment for patients with IPF who have discontinued

PFD treatment [12–15]; however, to the best of our knowledge, there are only a few reports on second-line PFD [16,17]. The change from PFD to NT has been reported in 10.5%—14.1% of patients [15–17]; however, in the present study, 53 of 170 patients (31.1%) were switched from NT to PFD because of unacceptable adverse effects (n = 25) and disease progression (n = 28). Three of 25 patients who were switched to PFD because of NT's adverse effects and one out of 28 patients who was switched to PFD because of disease progression discontinued treatment because of PFD's adverse effects. These results indicate that second-line PFD treatment was tolerated by most patients (90.6%). Patients who were switched to PFD because of NT's adverse effects had stable FVC values six months after the switch. Furthermore, after switching from NT to PFD because of disease progression, 19 patients had stable disease and nine others patients had worsening disease, and statistically significant inhibition of FVC decline was evident after switching to PFD treatment. Suzuki et al. reported that second-line anti-fibrotic treatment was well tolerated, especially in patients who changed their first-line drugs because of disease deterioration; however, in eight of the 19 patients who changed their drugs because of adverse effects, second-line anti-fibrotic treatment was discontinued, mainly because of gastrointestinal adverse effects [17]. On the other hand, Ikeda et al. reported that of the 30 patients with IPF who switched from PFD to NT, 18 (60%) discontinued treatment during NT treatment, mainly due to adverse effects including liver injury and anorexia with weight loss, and a further 16 (88.9%) of these patients had discontinued NT within six months [13]. Given the differences in important patient backgrounds (such as the IPF disease severity) between our report and previous reports, there are possible differences in the tolerability and efficacy results of anti-fibrotic drugs used as second-line treatments. Although previous reports have mainly focused on studies that entail switching from PFD to NT, we believe that PFD is well tolerated as a second-line treatment from first-line NT and that switching to PFD is more effective, in patients who change treatment due to disease progression than in those who change treatment due to adverse effects.

Nevertheless, our study had several limitations. First, the study had a retrospective single-centre design, and the sample size was too small to produce conclusive results. Second, there were no standards for the timing of anti-fibrotic drug changes or the management of adverse effects in this study, and there is a lack of objectivity. However, based on the findings of the study, it was possible to suggest the significance and necessity of changing anti-fibrotic drugs. Thus, larger prospective studies are required to address these limitations.

In conclusion, we demonstrated that introducing first-line NT treatment as early as possible in patients with IPF (regardless of their age) is associated with high continuity rates and effective conversion to PFD in the event of deterioration.

## Supporting information

**S1 Data.**
(XLS)

## Acknowledgments

We thank Chiaki Nishimura, Professor Emeritus of Toho University and Representative of CN Medical Research, for helpful discussions and statistical analyses. We also thank Dr. Akira Hebisawa (Department of Pathology, Tokyo National Hospital, Tokyo, Japan) for investigating histologically, and Dr. Atsuko Kurosaki (Department of Diagnostic Radiology, Fukujuji Hospital, Anti-tuberculosis Association, Tokyo, Japan) for the interpretation of radiological findings.

## Author Contributions

**Conceptualization:** Keishi Sugino, Hirotaka Ono, Eiyasu Tsuboi.

**Data curation:** Keishi Sugino, Hirotaka Ono, Mikako Saito, Masahiro Ando, Eiyasu Tsuboi.

**Formal analysis:** Keishi Sugino, Hirotaka Ono.

**Investigation:** Keishi Sugino, Hirotaka Ono, Mikako Saito, Eiyasu Tsuboi.

**Methodology:** Keishi Sugino.

**Validation:** Keishi Sugino.

**Writing – original draft:** Keishi Sugino.

**Writing – review & editing:** Hirotaka Ono, Mikako Saito, Masahiro Ando, Eiyasu Tsuboi.

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
