## [Decision Letter · Decision Letter 0]

8 May 2024

PONE-D-24-14937Tolerability and efficacy of switching anti-fibrotic treatment from nintedanib to pirfenidone for idiopathic pulmonary fibrosisPLOS ONE

Dear Dr. Sugino,

Thank you for submitting your manuscript to PLOS ONE. After careful consideration, we feel that it has merit but does not fully meet PLOS ONE’s publication criteria as it currently stands. Therefore, we invite you to submit a revised version of the manuscript that addresses the points raised during the review process.

We look forward to receiving your revised manuscript.

Kind regards,

Yoshiaki Zaizen, MD, PhD

Academic Editor

PLOS ONE

Journal Requirements:

2. Thank you for stating the following in the Competing Interests/Financial Disclosure * (delete as necessary) section:

"All authors contributed substantially to this work and are responsible for the consent of the manuscript.

KS have received lecture fees from Nippon Boehringer Ingelheim Co., Ltd and Shionogi & Co., Ltd.

The other authors have no financial relationships relevant to this article.

All authors have no competing non-financial interests."

We note that you received funding from a commercial source: "KS have received lecture fees from Nippon Boehringer Ingelheim Co., Ltd and Shionogi & Co., Ltd."

3. We note that your Data Availability Statement is currently as follows: "All relevant data are within the manuscript and its Supporting Information files."

Additional Editor Comments :

Thank you for submitting your manuscript to PLOS ONE. We kindly request that you resubmit your manuscript after appropriate response to the reviewer's comments.

Reviewers' comments:

Reviewer's Responses to Questions

**Comments to the Author**

1. Is the manuscript technically sound, and do the data support the conclusions?

Reviewer #1: Yes

Reviewer #2: Yes

2. Has the statistical analysis been performed appropriately and rigorously? 

Reviewer #1: Yes

Reviewer #2: Yes

3. Have the authors made all data underlying the findings in their manuscript fully available?

Reviewer #1: Yes

Reviewer #2: Yes

4. Is the manuscript presented in an intelligible fashion and written in standard English?

Reviewer #1: Yes

Reviewer #2: Yes

5. Review Comments to the Author

Reviewer #1: Author submitted an interesting manuscript about switching anti-fibrotics. I have several concerns for author.

1. You showed two patterns of elderly IPF patients in Table 3. Which threshold is better for considering elderly IPF patients based on your cohort ?

2. Regarding adverse effect of anti-fibrotics, we also consider the possibility of interaction of other medication especially in elderly patients with polypharmacy. How about the possibility of drug interaction of each case ?

3. When you commence PFD in your cohort, did you do any specific management especially in nintedanib intolerance patients not disease progression ?

4. Which kind of disease progression of your switching anti-fibrotics such as FVC decline with worsening symptoms, progression of HRCT imaging of fibrosis and DLco decline ? We would like to know more detailed information.

5. How about the smoking effect of discontinuation of anti-fibrotics ?

6. Secondary PH is also crucial comorbidity in IPF patients. Could you show us the incidence of PH of both continuation and discontinuation group ?

7. How about the change of body weight when did you switch anti-fibrotics ?

Reviewer #2: 1. General comment

This manuscript is well written and well presented.

2. Major comments

・In the present study, 72.4% continued nintedanib for 1 year. Real-world data showed that about 50% of Japanese IPF patients discontinued nintedanib treatment within the first year (Ref 23. Ogura et al.). Probably, IPF management in the author’s hospital have been well managed to prevent adverse events of nintedanib itself.

On the other hand, when it comes to deciding whether disease progression or adverse event (e.g., appetite loss, fatigue), it’s hard to know where to draw the line?

・A post-marketing surveillance study that included Japanese IPF patients treated with nintedanib showed that older age, male sex, small body surface area, severe IPF, and lower %FVC predicted at baseline were associated with an increased risk of early discontinuation of nintedanib treatment. The present study showed similar results (i.

e., lower BMI, and lower initial %FVC, higher GAP stage). However, no significant difference was observed in the nintedanib discontinuation rate between elderly and non-elderly patients. This is because many patients were preserved physique, not limited to mild IPF)

・Which reference do authors define disease progression? Please mention.

・In results paragraph, authors mentioned “Next, 80 patients (65%) had to reduce the NT dose because of the following reason:・・・”. During 1 year or whole follow-up period?

・Initial NT dosage in all patients was 150mg×2/day? During follow-up, the patients increasing the dosage halfway was none?

・In table 4, higher GAP stage was risk factors for discontinuing NT within 12 months. On the other hand, a lower BMI was most significant predictor of deterioration after switching from NT to PFD, not higher GAP stage. Although I predicted the similar results in the above two analyses, how should I interpret it?

6. PLOS authors have the option to publish the peer review history of their article (what does this mean?). If published, this will include your full peer review and any attached files.

Reviewer #1: No

Reviewer #2: No

---

## [Author Response · Author response to Decision Letter 0]

20 May 2024

Responses to Reviewer 1

Comments to the Author:

Reviewer #1: Author submitted an interesting manuscript about switching

anti-fibrotics. I have several concerns for author.

Response: We appreciate the reviewer’s kind comments. We are grateful to Reviewer 1 for the critical comments and useful suggestions that have helped us to improve our paper considerably. As indicated in the responses that follow, we have taken this comment and suggestions into account in the revised version of our paper.

1. You showed two patterns of elderly IPF patients in Table 3. Which

threshold is better for considering elderly IPF patients based on your

cohort?

Response: We are in agreement regarding clarifying this point. As mentioned in the discussion, because a significantly higher proportion of patients with early NT dose reduction within six months in the elderly group than in the non-elderly group, we think that the NT discontinuation rates did not differ significantly between elderly (≥75 years or ≥80 years) and non-elderly (<75 years or <80 years) patients with IPF. Therefore, In the aging society that we are facing today, we believe it is significant that the tolerability of NT was confirmed even in patients aged 75 or 80 years or older. In addition, since there are few studies of patients over 80 years of age, we have adopted both age thresholds and clearly stated them, without implying that one is better than the other.

2. Regarding adverse effect of anti-fibrotics, we also consider the

possibility of interaction of other medication especially in elderly

patients with polypharmacy. How about the possibility of drug

interaction of each case?

Response: Thank you for your important comments. We agree on this point regarding drug interactions in elderly patients with polypharmacy. We have treated our patients with close attention to the interactions of NT with the various medications they are already taking. 

3. When you commence PFD in your cohort, did you do any specific

management especially in nintedanib intolerance patients not disease

progression?

Response: Thank you for your critical comments. Consideration was given to confirming that NT side effects had reduced or disappeared, or that the drug was withdrawn for a while if there was no evidence of disease progression.

4. Which kind of disease progression of your switching anti-fibrotics

such as FVC decline with worsening symptoms, progression of HRCT imaging

of fibrosis and DLco decline? We would like to know more detailed

information.

Response: We understand the referee’s point. As you have pointed out, information relevant to this issue has been added in the revised manuscript.

Results: Efficacy of PFD in patients who switched from NT

The reasons for worsening after the switch were as follows: in 4 patients, %FVC and %DLco decreased, symptoms worsened, and imaging findings worsened; in 3 patients, %FVC decreased, symptoms worsened, and imaging findings worsened; and in 2 patients, %DLco decreased, symptoms worsened, and imaging findings worsened.

5. How about the smoking effect of discontinuation of anti-fibrotics?

Response: Thank you for your significant comments. A comparison between the NT continuation group and the noncontinuation group by smoking status (Current smoker/Former smoker/Never smoker) showed no statistically significant difference.

6. Secondary PH is also crucial comorbidity in IPF patients. Could you

show us the incidence of PH of both continuation and discontinuation

group?

Response: We understand the referee’s point. A comparison between the NT continuation group and the noncontinuation group by complication of secondary PH showed no statistically significant difference. As you have pointed out, information relevant to this issue has been added in the revised manuscript (Methods and Results; Table 1).

Methods: Doppler echocardiography

The estimated systolic pulmonary arterial pressure (esPAP) was calculated from measurements using transthoracic Doppler echocardiography by specific technicians. The transtricuspid pressure gradient was calculated using the modified Bernoulli equation and was considered to be equal to the equal to the esPAP in the absence of right ventricular outflow obstruction: esPAP = transtricuspid pressure gradient + right atrial pressure. Pulmonary arterial hypertension was defined as a esPAP >35 mmHg at rest.

7. How about the change of body weight when did you switch anti-

fibrotics?

Response: Thank you for your significant comments. The change in body weight was -0.8±3.8 kg in the 6-month period before the PFD change and -1.4±3.3 kg in the 6-month period after the PFD change. Thus, information relevant to this issue has been added in the revised manuscript (Methods and Results).

Methods: Statistical analysis: 

Mean changes in FVC and body weight values for six months before and after treatment with anti-fibrotic agents were compared using the one-way repeated-measure analysis of variance (ANOVA) with Bonferroni’s multiple comparison test.

Results: Efficacy of PFD in patients who switched from NT

Also, the weight change in the 6 months before PFD switch was -0.8±3.8 kg and in the 6 months after PFD switch was -1.4±3.3 kg, showing no difference in weight change before and after PFD switch (64.6 ± 10.3 kg, 63.7 ± 9.9 kg, 62.3 ± 11.2 kg; P = 0.69, P = 0.16).

 

Responses to Reviewer 2

Comments to the Author:

Reviewer #2: 1. General comment

This manuscript is well written and well presented.

Response: We appreciate the reviewer’s kind comments. We are grateful to Reviewer 2 for the critical comments and useful suggestions that have helped us to improve our paper considerably. As indicated in the responses that follow, we have taken this comment and suggestions into account in the revised version of our paper.

Major comments

1. In the present study, 72.4% continued nintedanib for 1 year. Real-

world data showed that about 50% of Japanese IPF patients discontinued

nintedanib treatment within the first year (Ref 23. Ogura et al.).

Probably, IPF management in the author’s hospital have been well managed

to prevent adverse events of nintedanib itself.

On the other hand, when it comes to deciding whether disease progression

oradverse event (e.g., appetite loss, fatigue), it’s hard to know where

to draw the line?

Response: Thank you for your kind comments. As you have pointed out, we agree that the decision to continue NT may be difficult to make based on evaluation of side effects and disease progression in real-world clinical setting. However, we believe that delaying these decisions may affect the patient's prognosis. Therefore, as in our study, we believe that it is important to reduce or stop the dose as an early countermeasure against adverse effects and decide whether to continue or change the drug according to the aggravation criteria.

2. A post-marketing surveillance study that included Japanese IPF

patients treated with nintedanib showed that older age, male sex, small

body surface area, severe IPF, and lower %FVC predicted at baseline were

associated with an increased risk of early discontinuation of nintedanib

treatment. The present study showed similar results (i.e., lower BMI, and lower initial %FVC, higher GAP stage). However, no significant difference was observed in the nintedanib discontinuation rate between elderly and non-elderly patients. This is because many patients were preserved physique, not limited to mild IPF.

Response: Thank you for your significant comments. We agree on this point. Thus, we have amended in discussion of the revised manuscript.

A post-marketing surveillance study of Japanese IPF patients treated with nintedanib showed that older age (≥75 years), male age, smaller body surface area (1.58 m2), severe IPF, and lower predicted %FVC (< 70%) at baseline were associated with an increased risk of early discontinuation of treatment with NT [26]. In our study, the NT discontinuation rates did not differ significantly between elderly (≥75 years or ≥80 years) and non-elderly (<75 years or <80 years) patients with IPF. Risk factors for discontinuation within 12 months were higher GAP severity, but older age was not a significant factor. We believe this was partly due to the subjective judgement of the attending physician regarding the elderly. As a result, the proportion of patients who reduced their NT dose earlier within 6 months was significantly higher in elderly patients than in the non-elderly patients. In addition, this may be due to the fact that the %FVC (≥75 years: 76.6±17.9%, ≥80 years: 77.8±17.6%), %DLco (≥75 years: 73.9±23.8%, ≥80 years: 74.2±25.9%), and BMI (≥75 years: 22.6±3.9%, ≥80 years: 22.1±4.4%) were relatively preserved in elderly patients.

3. Which reference do authors define disease progression? Please mention.

Response: We understand the referee’s point. As you have pointed out, information relevant to this issue has been added in the revised manuscript.

To evaluate treatment response, we defined disease progression that met at least one of the following progression criteria within six months: i) a relative decline of ≥10% of the predicted FVC (%FVC); ii) a relative decline of ≥15% of the predicted DLco (%DLco); iii) a relative decline of ≥5% but <10% decline in %FVC along with increasing fibrosis in chest CT images; iv) a relative decline of ≥5% but <10% decline in %FVC along with deteriorating respiratory symptoms [20, 21].

References: 

20. Flaherty KR, Wells AU, Cottin V, Devaraj A, Walsh SLF, Inoue Y, et al. Nintedanib in Progressive Fibrosing Interstitial Lung Diseases. N Engl J Med 2019; 381: 1718-1727.

21. George PM, Spagnolo P, Kreuter M, Altinisik G, Bonifazi M, Martinez FJ, et al. Progressive fibrosing interstitial lung disease: clinical uncertainties, consensus recommendations, and research priorities. Lancet Respir Med. 2020 Sep; 8(9): 925-934.

4. In results paragraph, authors mentioned “Next, 80 patients (65%) had

to reduce the NT dose because of the following reason:・・・”. During 1

year or whole follow-upperiod?

Response: We apologize for confusing you about suggested sentences. We have changed this information in the revised manuscript.

Results: 

Next, 80 patients (65%) had to reduce the NT dose during 12 months

5. Initial NT dosage in all patients was 150mg×2/day? During follow-up,

the patients increasing the dosage halfway was none?

Response: We understand the referee’s point. All patients with IPF in our hospital have been started on treatment with NT at an initial dose of 300 mg.

Therefore, we have amended in methods of the revised manuscript.

Methods:

All patients with IPF in our hospital have been started on treatment with NT at an initial dose of 300 mg.

6. In table 4, higher GAP stage was risk factors for discontinuing NT

within 12 months. On the other hand, a lower BMI was most significant

predictor of deterioration after switching from NT to PFD, not higher

GAP stage. Although I predicted the similar results in the above two

analyses, how should I interpret it?

Response: We are in agreement regarding clarifying this point. 

In the analysis of risk factors for discontinuation of NT within 12 months and predictors of deterioration after switching from NT, different factors were identified in the multivariate analysis, but in the univariate analysis, low BMI was identified as a factor in two analyses, which is understandable.

---

## [Decision Letter · Decision Letter 1]

30 May 2024

Tolerability and efficacy of switching anti-fibrotic treatment from nintedanib to pirfenidone for idiopathic pulmonary fibrosis

PONE-D-24-14937R1

Dear Dr. Sugino,

We’re pleased to inform you that your manuscript has been judged scientifically suitable for publication and will be formally accepted for publication once it meets all outstanding technical requirements.

Kind regards,

Yoshiaki Zaizen, MD, PhD

Academic Editor

PLOS ONE

Additional Editor Comments (optional):

Reviewers' comments:

Reviewer's Responses to Questions

**Comments to the Author**

1. If the authors have adequately addressed your comments raised in a previous round of review and you feel that this manuscript is now acceptable for publication, you may indicate that here to bypass the “Comments to the Author” section, enter your conflict of interest statement in the “Confidential to Editor” section, and submit your "Accept" recommendation.

Reviewer #1: All comments have been addressed

Reviewer #2: All comments have been addressed

2. Is the manuscript technically sound, and do the data support the conclusions?

Reviewer #1: Yes

Reviewer #2: Yes

3. Has the statistical analysis been performed appropriately and rigorously? 

Reviewer #1: Yes

Reviewer #2: Yes

4. Have the authors made all data underlying the findings in their manuscript fully available?

Reviewer #1: Yes

Reviewer #2: Yes

5. Is the manuscript presented in an intelligible fashion and written in standard English?

Reviewer #1: Yes

Reviewer #2: Yes

6. Review Comments to the Author

Reviewer #1: Author responded my questions properly. It is informative for chest physician. Current status will be acceptable for publication.

Reviewer #2: The authors revised the manuscript to address almost my questions and concerns. I think the revised manuscript has been properly fixed.

7. PLOS authors have the option to publish the peer review history of their article (what does this mean?). If published, this will include your full peer review and any attached files.

Reviewer #1: No

Reviewer #2: No

---

## [Editor Report · Acceptance letter]

4 Jun 2024

PONE-D-24-14937R1 

PLOS ONE

Dear Dr. Sugino, 

I'm pleased to inform you that your manuscript has been deemed suitable for publication in PLOS ONE. Congratulations! Your manuscript is now being handed over to our production team.

Kind regards, 

on behalf of

Dr. Yoshiaki Zaizen 

Academic Editor

PLOS ONE